

# Global TEC prediction performance assessment of IRI-2016 model based on EOF decomposition

Shuhui Li[1,2], Jiajia Xu[1], Houxiang Zhou[1], Jinglei Zhang[1], Zeyuan Xu[1], Mingqiang Xie[1]

[1] School of Land Science and Technology, China University of Geosciences (Beijing), Beijing 100083, China
[2] Shanxi Key Laboratory of Resources, Environment and Disaster Monitoring, Jinzhong 221116, China

*Correspondence to*: Shuhui Li (li.shuhui@163.com)

**Abstract:** In this study, the empirical orthogonal function (EOF) decomposition technique was utilized to analyze the similarities and differences of the spatiotemporal characteristics between the total electron content (TEC) of the International GNSS Service global ionospheric map (GIM) and that derived from the International Reference Ionosphere 2016 (IRI-2016) model in 2013. Results showed that the main spatial patterns and time-varying features of the data set have good consistency. The following four main spatiotemporal variation features can be extracted from both data sets through EOF decomposition: the variation with the geomagnetic latitude reflecting the daily averaged solar forcing, the diurnal and semidiurnal periodic changes with longitude due to local time, and the interhemispheric asymmetry caused by the annual variation of the inclination angle of the Earth's orbit. The differences between the spatial patterns represented by the EOF base functions of IRI-2016 and GIM TECs were analyzed by extracting the same time-varying coefficients. The deviations of the interhemispheric asymmetry component between the two data sets showed roughly equal values throughout the Southern or Northern Hemisphere, whereas those of the other spatial modes were mainly concentrated on the equatorial region. The differences of the time-varying characteristics between the IRI-2016 and GIM TECs were also compared by extracting the same EOF base functions. Although the EOF coefficients of the two data sets presented consistent seasonal variations, the magnitude of IRI-2016 TEC changes over time was less than that of GIM TEC. The diurnal variation of the daily averaged solar forcing component and the annual variation of the interhemispheric asymmetry component exhibited relatively large deviations between the two data sets. Considering the variance contribution of the different EOF components and their average relative deviations, both analyses showed that the daily averaged solar forcing and interhemispheric asymmetry components were the main factors for the deviation between the IRI-2016 and GIM TECs.

## 1 Introduction

The ionosphere is a shell of electrons and electrically charged atoms and molecules that surrounds the Earth and stretches from a height of approximately 60 km to more than 1000 km. The variations in the ionosphere should be accurately measured, modeled, or estimated because the ionosphere critically affects high-frequency satellite communication and navigation system signals. Total electron content (TEC), which is the number of free electrons along the path where the signal is traveling, is a critical quantity that describes the ionosphere and its variability. Modeling and predicting temporal and spatial variations in ionospheric TEC are crucial to ionospheric physics research and ionospheric-based applications(Yao et al., 2018).

Many attempts have been made to specify ionospheric parameters using empirical approaches, because an empirical model can describe the general condition of the ionosphere without actual measured data (Feltens et al., 2011). Several ionosphere empirical models, such as Klobuchar, NeQuick, Standard Plasmasphere Ionosphere Model(SPIM), and International Reference Ionosphere (IRI; Bilitza 2001), are currently available. The IRI is one of the most accepted standard global empirical ionosphere models among others. This model can be used to estimate the values of electron density and temperature, ion temperature and composition, and TEC at altitudes ranging from approximately 50 km to 2000 km at a particular location, time, and day. The IRI model is continuously improved when new data and techniques become available. This model was recently upgraded to the IRI-



2016 version (Bilitza et al., 2017). The model has been improved by ingesting all available data from worldwide ground-based and satellite observations to enhance the model capacity. IRI-2016 includes two new model options for the F2 peak height hmF2 and an enhanced representation of topside ion densities at low and high solar activities. Several small changes were made concerning the use of solar indices and the speedup of the computer program (Bilitza et al., 2017).

The performance of the previous versions of the IRI model in terms of predicting TEC have been investigated to improve the model effectively and provide reference for the application (Maltseva et al., 2012; Scidá et al., 2012; Li et al., 2013; Kenpankho et al., 2013; Okoh et al., 2013; Zakharenkova et al., 2015). Comparative studies with GNSS-derived TEC have validated the performance of different IRI versions over years of varied solar activity in diverse regions. Given the predictability of the diurnal variation of TEC, deficiencies have varied with local time (LT), season, and latitude. After the release of IRI-2016 as the recent
version, its performance in predicting TEC has attracted the attention of many researchers (Atici, 2018; Sharma et al.,2018; Tariku, 2018; Jiang et al., 2018). Most existing studies for ionospheric models aimed at the low and middle latitudes. Studies on the TEC prediction performance of different IRI versions worldwide are relatively sparse. Most comparative studies are based on the contrast of the IRI model and global ionospheric map (GIM)- or GNSS-derived TEC. The variations of diurnal and seasonal changes and those in different solar activity years on certain sites have been investigated from several aspects, such as bias, root
mean square (RMS) error, and correlation coefficients. Although several assessments of the IRI models have been conducted, few studies on the comprehensive evaluation of the temporal and spatial distribution prediction performance of the IRI model are available. The predictive performance of the IRI model for ionospheric temporal and spatial changes should be evaluated using efficient analytical methods.

Many scholars have recently used the empirical orthogonal function (EOF) decomposition method to analyze the spatial patterns
and time temporal variations of the TEC and their relationships with influencing factors (Zhao et al., 2005; Mao et al., 2008; Zhang et al., 2011; Bouya et al., 2012; Zhang et al., 2013; Uwamahoro and Habarulema, 2015; Talaat and Zhu, 2016; Dabbakuti and Ratnam, 2016, 2017; Chang et al., 2017; Andima et al., 2019; Li et al., 2019). The spatial patterns and temporal variations of the TEC are separated by EOF decomposition and can be properly represented by the base functions and associated coefficients, respectively. The data analysis results of a single station and the regional or global TEC indicated that the EOF method is a
potentially useful tool for data compression and separation of different physical processes. The EOF method contributes to the comprehensive analysis of the overall spatiotemporal variations in ionospheric TEC.

In this work, GIM TEC data in 2013 were selected as reference values, and the EOF method was introduced to analyze the global TEC prediction performance of IRI-2016. A comparison between the modeled TEC and the reference values was conducted from the perspective of spatial patterns and time variation characteristics. Results provide a reference for the further understanding of
the differences between the IRI-2016 and the GIM TECs at a global scale.

## 2 Data and method

### 2.1 GIM TEC

The GIM TEC used in this study is the official IGS combined final product provided by the Crustal Dynamic Data Information System (ftp://cddis.gsfc.nasa.gov). GIMs are regular products of the International GNSS Service (IGS) since 1998. These GIMs
are provided in the ionosphere exchange format with a spatial resolution of 2.5 °×5 ° in geographic latitude and longitude and a temporal resolution of 15 min to 2 h.

In this study, we downloaded and extracted the 2013 global TEC data from GIMs (referred to as GIM-TEC hereafter).





## 2.2 IRI-2016

The IRI is the international standard empirical model for terrestrial ionosphere and recommended for international use by the Committee On Space Research and International Union of Radio Science (Bilitza, 2001; Bilitza and Reinisch, 2008; Chauhan and Singh, 2010). The first version was released in 1978, followed by several steadily improved ones in 1986, 1990, 1995, and

2012 (Rawer et al., 1978; Bilitza, 2015). The recent version of this model is IRI-2016 (Bilitza et al., 2016; Bilitza et al., 2017). After IRI-2012, IRI-2016 exhibits the latest improvement in the model by introducing two new F2 peak height hmF2 modeling options with their data sources from ionosonde measurements and COSMIC radio occultations. Hence, this version is independent of the propagation factor M(3000)F2 (Bilitza et al., 2017).

The software package of IRI-2016 can be downloaded from http://irimodel.org/. The IRI software package contains FORTRAN

subroutines, model coefficients, index files for IRI-2016 models, README files, and license files. The user can calculate relevant parameters by inputting location, time, height range, model selection, and certain parameters. The global TEC date calculated by using IRI-2016 will be called IRI-TEC hereafter. IRI-TEC can also be calculated online in accordance with http://omniweb.gsfc.nasa.gov/vitmo /iri2016_vitmo.html.

## 2.3 EOF decomposition

The EOF decomposition analysis method was originally invented by Pearson (1901). This method is performed by using an orthogonal transformation to decompose the original data set into a set of uncorrelated and ordered base functions and associated coefficients.

If an original data matrix $X$ with the dimension M×N is present, then the covariance matrix is determined from the data matrix $X$ in accordance with

$$\Sigma = X^T X .\tag{1}$$

The EOF base functions $E_i$, with i = 1, 2, 3,…, N, are the eigenvectors of the covariance matrix and obtained by solving

$$\Sigma E_i = \lambda_i E_i ,\tag{2}$$

where $\lambda_i$ is the associated eigenvalues. Once the EOF base functions are known, the EOF coefficients $A_k$ are obtained using

$$A_k = X E_k .\tag{3}$$

The original data set $X$ can be decomposed in terms of the EOF base functions and associated coefficients in accordance with

$$X = \sum_{k=1}^{N} A_k E_k .\tag{4}$$

The percentage of the total variance in the data set accounted for by the $i$ th EOF component is given as follows:

$$r_i = 100 \times \frac{\lambda_i}{\sum_{j=1}^{N} \lambda_j} \% ,\tag{5}$$

where $N$ denotes the total number of the EOF components accounting for the total variance in the original data set.

Talaat and Zhu (2016) reported that the effectiveness of the EOF technique for TEC is nearly insensitive to the horizontal resolution and length of the data records. We analyzed the global TEC over a 1 year time period (2013) with a 2 h temporal resolution and 37×36 spatial grids.

We first organized the data set $TEC(Lat, Lon, UT, Doy)$ used in this study into a 2D matrix according to location and time epoch, that is, $TEC(epoch, grid)$, where $grid$ is a grid point arranged according to the latitude and longitude, and its total number is

37×36=1332; and $epoch$ is arranged according to University Time (UT), with an interval of 2 h. The total epoch number of the





study period was $12\times365=4380$. After performing EOF decomposition, the base function $E_k(grid)$ expressing a spatial pattern and the associated coefficient $A_k(epoch)$ varying with time are obtained.

The EOF method can separate the temporal and spatial variation characteristics. If IRI-TEC and GIM-TEC data are decomposed, then their EOF base functions and coefficients will exhibit poor comparability. Therefore, we combined the data to form a whole

data set for EOF decomposition and compared the two data sets.

The same coefficients of the EOF base function, that is, the same time-varying features, can be obtained by arranging IRI-TEC and GIM-TEC according to the same number of columns. Accordingly, comparing the two modes' spatial variation features represented by the base functions is feasible.

$$\begin{bmatrix} X_{GIM} \\ X_{IRI} \end{bmatrix} = \sum_{k=1}^{N} A_k \cdot \begin{bmatrix} E_{k,GIM} \\ E_{k,IRI} \end{bmatrix} \qquad (6)$$

If IRI-TEC and GIM-TEC are arranged in the same number of rows, then the same spatial variation features represented by EOF base functions will be obtained. Accordingly, the time variation characteristics of the two data sets can be compared.

$$[X_{GIM} \quad X_{IRI}] = \sum_{k=1}^{N} \begin{bmatrix} A_{k,GIM} & A_{k,IRI} \end{bmatrix} \cdot E_k \qquad (7)$$

### 2.4 Evaluation indicators

In this study, the mean bias was calculated to represent the difference between two data sets. The equation is shown as follows:

$$Bias = \frac{1}{n}\sum_{i=1}^{n}(Y_i - Y_i'), \qquad (8)$$

where $n$ is the total number of sample data, and $Y_i$ and $Y_i'$ are sample data for two different data sets. These variables can be TEC from IRI-2016 and GIMs or the values of base functions or coefficients of base functions. The mean relative bias (Bias_rel) can be calculated as follows:

$$Bias\_rel\% = \frac{1}{n}\sum_{i=1}^{n}\frac{(Y_i - Y_i')}{Y_i'}\times100. \qquad (9)$$

The RMS of the bias can be calculated using the following expression:

$$RMS = \left[\sum_{i}^{n}(Y_i - Y_i')^2 / n\right]^{1/2}. \qquad (10)$$

The 2D linear correlation coefficient was used to investigate the similarity of the spatial pattern of IRI-TEC and GIM-TEC. The 2D linear correlation coefficient $\rho$ for two matrices $A$ and $B$ with $M\times N$ dimension is calculated as

$$\rho = \sum_{m=1}^{M}\sum_{N=1}^{N}(A_{mn} - \overline{A})(B_{mn} - \overline{B})\cdot\left[\sum_{m=1}^{M}\sum_{N=1}^{N}(A_{mn} - \overline{A})^2\right]^{-\frac{1}{2}}\cdot\left[\sum_{m=1}^{M}\sum_{N=1}^{N}(B_{mn} - \overline{B})^2\right]^{-\frac{1}{2}}, \qquad (11)$$

where $\overline{A}$ and $\overline{B}$ are the mean values of matrices $A$ and $B$, respectively, and they are written as

$$\overline{A} = \frac{1}{MN}\sum_{m=1}^{M}\sum_{n=1}^{N}A_{mn} \; ; \text{ and } \; \overline{B} = \frac{1}{MN}\sum_{m=1}^{M}\sum_{n=1}^{N}B_{mn} \; . \qquad (12)$$





## 3 Results and analysis

### 3.1 GIM-TEC and IRI-TEC in 2013

Figure 1 shows the season averages of global GIM-TEC and IRI-TEC at UT 12:00 in 2013. The months are divided into the following four seasons: March equinox (February, March, and April), June solstice (May, June, and July), September equinox

(August, September, and October), and December solstice (November, December, and January). The global level of ionospheric TEC at UT 12:00 is lowest during the June solstice compared with that during other seasons. By contrast, the ionospheric TEC reaches the highest level during the December solstice.

The figure illustrates that the spatial distribution characteristics, which change with the latitude and longitude exhibited by IRI-TEC and GIM-TEC, have good consistency. However, the equatorial ionospheric anomaly of IRI-TEC is more pronounced than

that of GIM-TEC. The 2D correlation coefficients of the two types of TEC data are shown in Table 1. The correlation coefficients of the four seasons are at least 0.924.

Table 1 reveals that the mean biases between the season averages of global IRI-TEC and GIM-TEC at UT 12:00 are all negative. This result indicates that the TEC level predicted by the IRI-2016 model is lower than that of the GIM. This characteristic can also be seen in Figure 1. The mean bias, and mean relative bias between IRI-TEC and GIM-TEC during the December solstice

are larger than those in other seasons.

The bias values between the IRI-TEC and GIM-TEC of all global grid points at different UTs were used to calculate the daily RMS in 2013. Results are shown in Figure 2, which also displays the daily solar F10.7 index and daily average of geomagnetic AE index in 2013. The solar F10.7 and geomagnetic AE indexes are available at https://omniweb.gsfc.nasa.gov/form/dx1.html.

Figure 2 demonstrates that the daily predicted RMS of IRI-2016 is in good agreement with the daily solar F10.7 index. The

correlation coefficients between the RMS and the solar F10.7 or geomagnetic AE index are 0.78 and −0.19, respectively. Results indicate that the ionospheric TEC prediction error of the IRI-2016 model presents a strong correlation with solar activity.

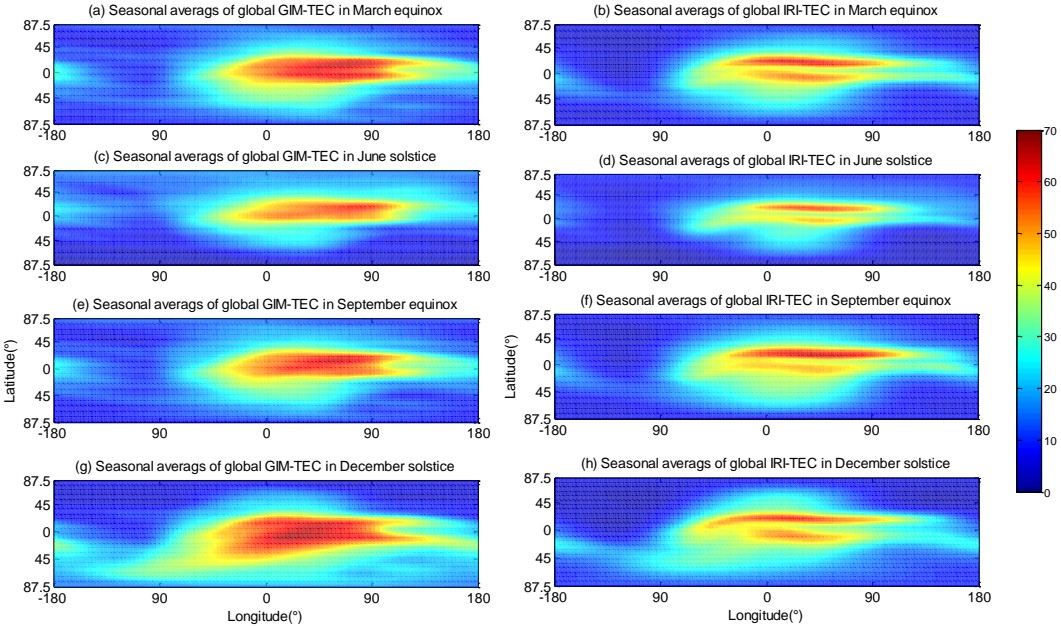

Figure 1. Season averages of global TEC obtained from GIM and IRI at UT 12:00 in 2013. (a) GIM-TEC in the March equinox; (b) IRI-TEC in the March equinox; (c) GIM-TEC in the June solstice; (d) IRI-TEC in the June solstice; (e) GIM-TEC in the





September equinox; (f) IRI-TEC in the September equinox; (g) GIM-TEC in the December solstice; and (h) IRI-TEC in the December solstice.

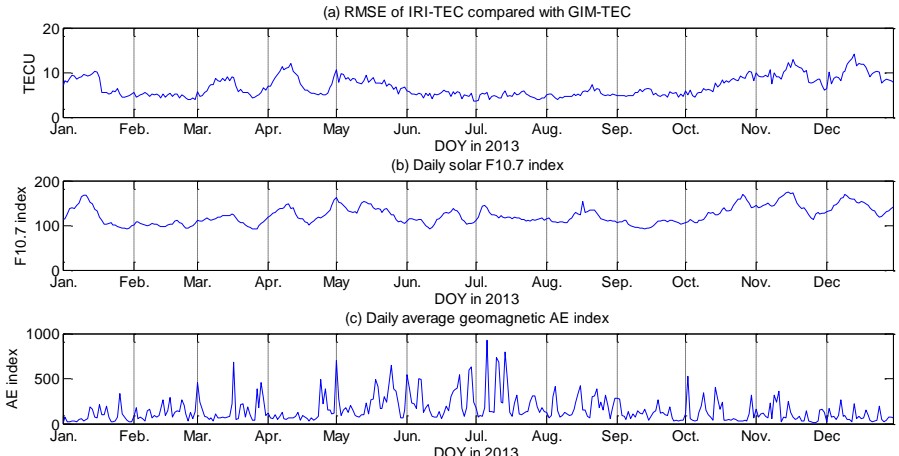

Figure 2. Daily (a) RMSE of IRI-TEC, (b) solar F10.7 index, and daily (c) average geomagnetic AE index in 2013.

5    Table 1. Correlation coefficient and bias statistics among the season averages of global IRI-TEC and GIM-TEC at UT 12:00 in 2013

|  | Correlation coefficient $\rho$ | Maximum bias | Minimum bias | Mean bias | Mean relative bias Bias_rel% |
|---|---|---|---|---|---|
| March equinox | 0.944 | 16.199 | −23.332 | −3.456 | −20.0% |
| June solstice | 0.948 | 7.7401 | −20.478 | −3.7193 | −19.8% |
| September equinox | 0.953 | 12.476 | −20.525 | −1.569 | −11.0% |
| December solstice | 0.924 | 14.866 | −27.728 | −5.743 | −23.1% |

**3.2 Differences of spatial patterns between IRI-TEC and GIM-TEC based on the same time-varying characteristics**

We combined the IRI-TEC and GIM-TEC data to obtain the same TEC time-varying characteristics using Eq. (6) and analyzed their differences in terms of spatial patterns.

10   The time-varying characteristics are reflected in the coefficient $A_k$ of the EOF decomposition. Given that the TEC data are in accordance with the 2 h time interval, coefficient $A_k$ is also the data that vary with the 2 h time interval. We described the coefficients of the base function according to the changes in UT and day of year (DOY) in Figure 3 to reflect the seasonal changes effectively.





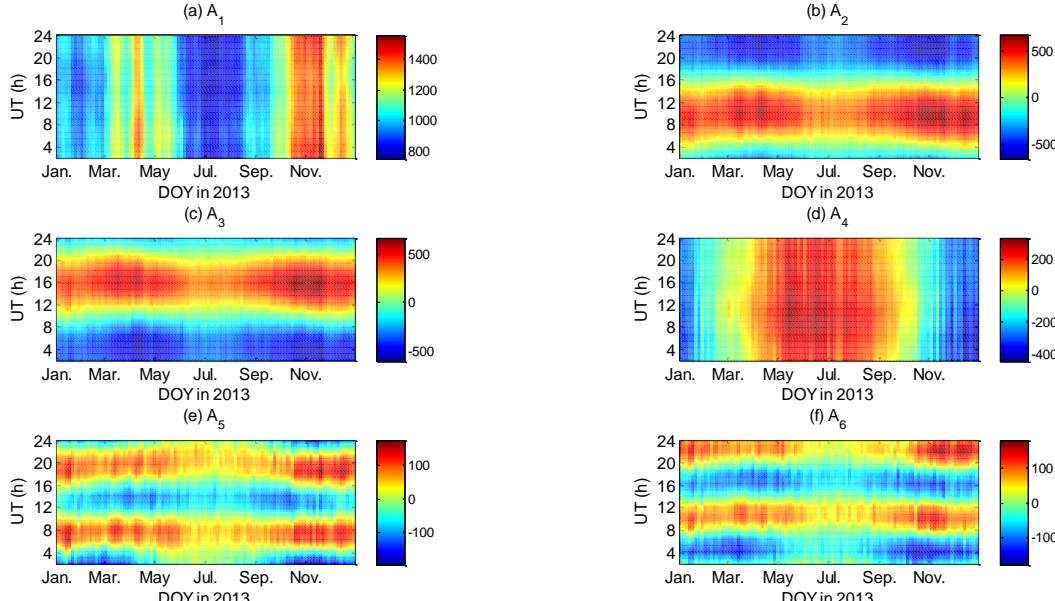

Figure 3. Associated coefficients $A_1 - A_6$ of the first six orders of EOF base functions based on Eq. (6), and $A_1 - A_6$ were plotted against UT and DOY.

The main EOF base functions extracted from Eq. (6) are shown in Figure 4. The graphics in the left column of Figure 4 exhibit the first six base functions $E_i$ of GIM-TEC, whereas those in the right column of Figure 4 depict the base functions of IRI-TEC.





Figure 4. First six orders of EOF base functions $E_1 - E_6$ extracted on the basis of Eq. (6). The figures in the left column are the

5    base functions of GIM-TEC, and those in the right column are the base functions of IRI-TEC.



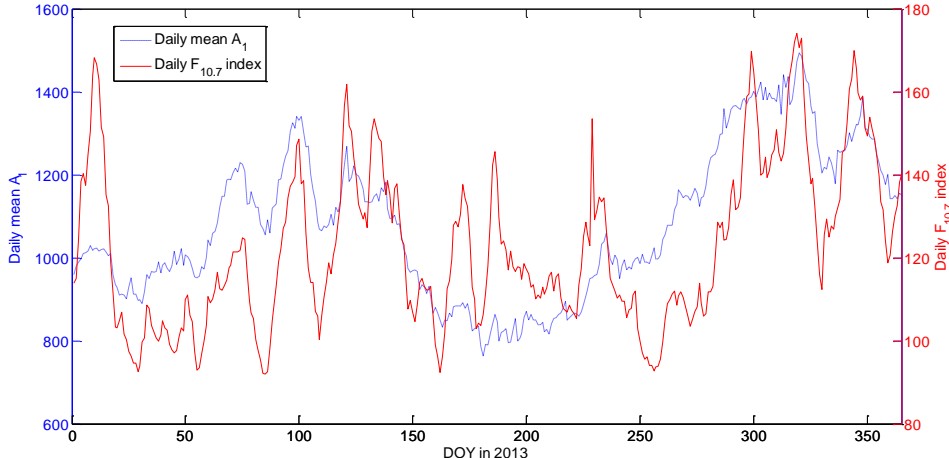

Figure 5. Daily mean first EOF coefficient $A_1$ and daily solar F10.7 index.

The first base function $E_1$ of GIM-TEC and IRI-TEC in Figs. 4(a) and (b) describe the overall average of global TEC. This function reflects the daily average effect of solar forcing and offset magnetic field (Talaat and Zhu, 2016). The TEC over the area near the geomagnetic equator exhibits a peak value. The TEC value decreases with the increase in geomagnetic latitude. The spatial distribution characteristics of $E_1$ of the two models are very consistent. However, the peak GIM-TEC value over the geomagnetic equator is greater than that of the IRI-TEC. The ionospheric trough near the geomagnetic equator is evident in Figure 4(b). The daily mean $A_1$ and solar F10.7 index are illustrated in Figure 5, which shows that these two data sets demonstrate a consistent trend. The correlation coefficient between daily mean $A_1$ and F10.7 index is 0.61. Solar activity is the primary determinant of the first base function $E_1$.

Figs. 4(c)–(f) present that the second and third base functions reflect the spatial distribution that varies along the longitude direction. The two base functions $E_2$ and $E_3$ approximately have the same magnitude and show a phase shift of $\pi/2$, which is consistent with the results of Talaat and Zhu (2016). These functions reflect the change of diurnal solar radiation as it changes with the LT. This change of GIM-TEC and IRI-TEC is generally consistent; their main difference is reflected in the peak region of the equator, and GIM-TEC shows large peak values. The EOF coefficients $A_2$ and $A_3$ corresponding to Figs. 3(b) and (c) show the change of the diurnal variation, and a change characteristic of the semiannual period is observed. The levels of $A_2$ and $A_3$ during equinox seasons are larger than those during solstice seasons.

The fourth base function $E_4$ reflects interhemispheric asymmetry, which is mainly caused by the seasonal variation of the inclination angle of the Earth's orbit. $A_4$ in Figure 3(d) indicates the seasonal variation of the interhemispheric asymmetry of the TEC and a strong annual cycle. The TEC component corresponding to base function $E_4$ in the Southern Hemisphere is positive. In the Northern Hemisphere, the maximum value of the $E_4$ component is on DOY150, whereas that in the Southern Hemisphere is on DOY347.

Similar to $E_2$ and $E_3$, the fifth and sixth base functions $E_5$ and $E_6$ also reflect the spatial distribution characteristics along the longitude (Figs. 4(i) to (l)). In conjunction with Figs. 3(e) and (f), these two base functions have semidiurnal period changes, and the phases of the two base functions differ by $\pi/4$ and are of approximately equal magnitude. Base functions $E_5$ and $E_6$





represent a semidiurnal variation that changes with LT, and their coefficients $A_5$ and $A_6$ show a semiannual period. The intensity of the semidiurnal variation is strong during the equinox season and weak during the June solstice.

We calculated the variances, correlation coefficients, biases, and their relative biases to analyze the spatial distribution characteristics of GIM-TEC and IRI-TEC. The statistical results are shown in Table 2, which indicates that the base functions of

the two data sets are correlated and present good consistency with Figure 4.

Table 2. Variances of the base function, correlation coefficient, and bias statistics among the base functions of GIM-TEC and IRI-TEC

| Base function | $E_1$ | $E_2$ | $E_3$ | $E_4$ | $E_5$ | $E_6$ |
|---|---|---|---|---|---|---|
| Variances $r_i$ | 79.03% | 8.24% | 7.52% | 2.55% | 0.37% | 0.35 |
| Correlation coefficient $\rho$ | 0.971 | 0.960 | 0.956 | 0.936 | 0.739 | 0.716 |
| Maximum bias | 0.0022 | 0.0189 | 0.0192 | 0.0276 | 0.0481 | 0.0587 |
| Minimum bias | −0.0105 | −0.0217 | −0.0243 | −0.0300 | −0.0528 | −0.0586 |
| Mean bias | −0.0035 | −0.00092 | −0.00056 | 0.00095 | −0.00593 | 0.00068 |
| Mean relative Bias Bias_rel% | −20.8% | −12.3% | −4.2% | −56.7% | −34.4% | −18.2% |

We showed the difference between the six base functions of GIM-TEC and IRI-TEC in Figure 6 to have an intuitive understanding of the difference between the IRI and the GIM base functions.

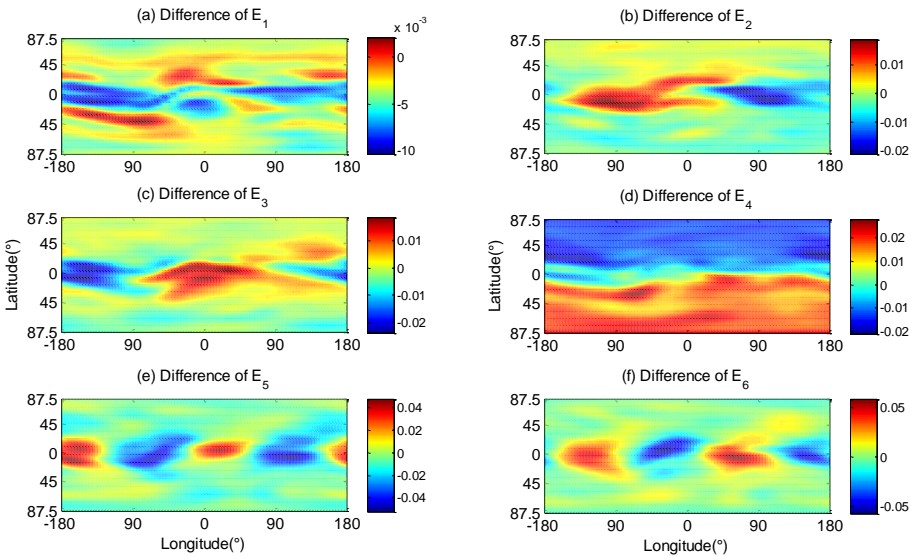

Figure 6. Differences of the first six orders of the base functions of GIM-TEC and IRI-TEC.

Figure 6 shows that the differences of other modes exhibit a large deviation in the equatorial and low latitude regions, except for the interhemispheric asymmetry feature $E_4$. The magnitudes of the spatial distribution changes of the IRI-TEC for all six base functions are significantly smaller than those of GIM-TEC.

The mean relative bias statistics of the base functions of GIM-TEC and IRI-TEC in Table 2 are negative. This finding indicates that the spatial variations of the base functions of IRI-TEC are generally underestimated compared with those of GIM-TEC. Here, the mean relative bias of $E_4$ reached −56%, and the underestimation is serious. This outcome is consistent with the statistical results in Table 1.



**3.3 Differences of time-varying characteristics between IRI-TEC and GIM-TEC based on the same spatial patterns**

Eq. (7) shows that the same EOF base functions are extracted for GIM-TEC and IRI-TEC. The differences of the corresponding coefficients of the EOF base functions between GIM-TEC and IRI-TEC are then compared, and those of their time variation characteristics can be analyzed.

Figure 7 shows the six EOF base functions extracted in accordance with Eq. (7). Similar to the EOF base function extracted in Figure 4, the first base function is consistent with the average variation of the TEC, varying with geomagnetic latitude. The second and third base functions are related to the diurnal variation of solar radiation change with longitude due to the LT. The fourth base function reflects the interhemispheric asymmetry caused by the seasonal variation of the inclination angle of the Earth's orbit. The fifth and sixth base functions reflect the characteristics of the semidiurnal variation with longitude due to the

LT.

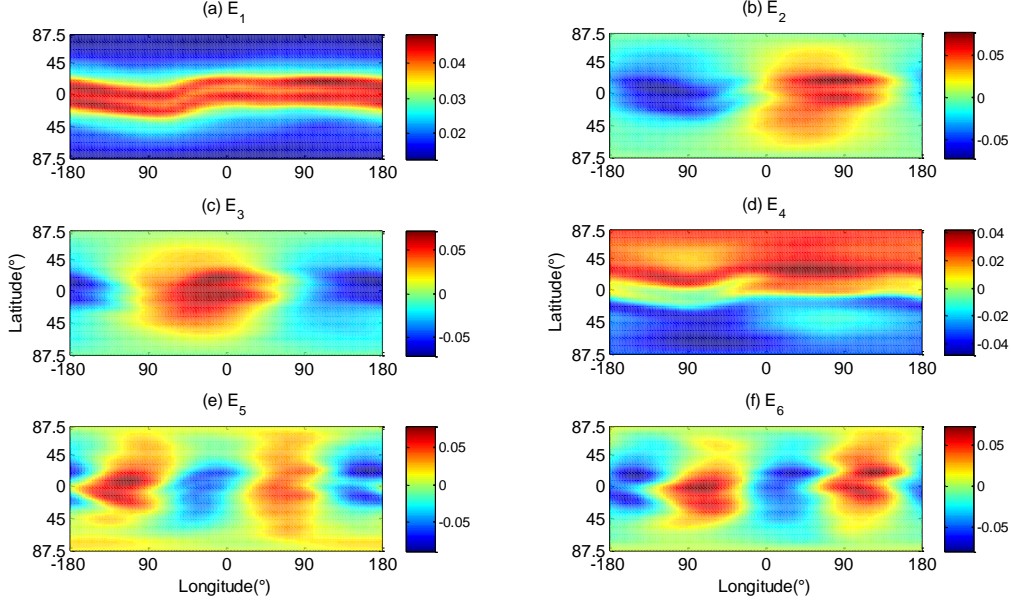

Figure 7. Six EOF base functions $E_1 - E_6$ extracted in accordance with Eq. (7).

The coefficients of the different base functions of GIM-TEC and IRI-TEC obtained in accordance with Eq. (7) are shown in Figure 8.

The figure manifests that the two data sets are highly consistent with the time variation of DOY and UT based on the same spatial distribution characteristics. The two sets of EOF coefficients have consistent annual, semiannual, diurnal, and semidiurnal changes. The variance and correlation coefficients of $A_1 - A_6$ of the two types of data and the bias statistics of such coefficients are shown in Table 3.



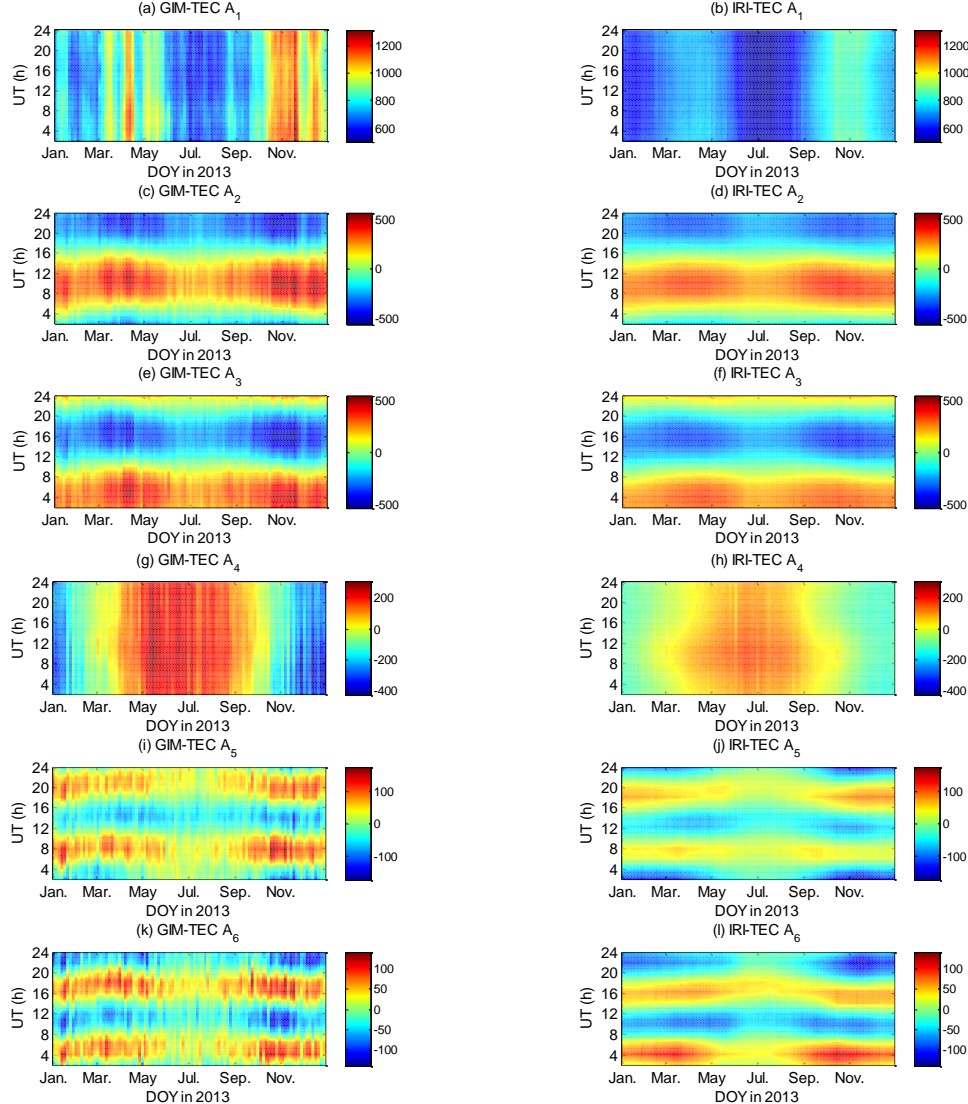

Figure 8. Associated coefficients $A_1 - A_6$ of the EOF base functions extracted in accordance with Eq. (7).

Table 3. Variances of base function, correlation coefficient, and bias statistics among coefficients $A_1 - A_6$ of GIM-TEC and IRI-

5    TEC

| Coefficient | $A_1$ | $A_2$ | $A_3$ | $A_4$ | $A_5$ | $A_6$ |
|---|---|---|---|---|---|---|
| Variances of base function $r_i$ | 79.03% | 8.24% | 7.52% | 2.55% | 0.37% | 0.35 |
| Correlation coefficient $\rho$ | 0.806 | 0.974 | 0.972 | 0.949 | 0.634 | 0.725 |
| Maximum bias | 118.24 | 252.39 | 246.44 | 323.55 | 112.84 | 143.39 |
| Minimum bias | −465.34 | −204.87 | −224.75 | −222.50 | −165.46 | −101.54 |
| Mean bias | −129.78 | 8.33 | 13.53 | −25.43 | −26.89 | 2.98 |
| Mean relative bias Bias_rel% | −16.94% | −10.62% | −10.98% | −52.83% | −38.82% | −17.98% |





The magnitudes of coefficients $A_1 - A_6$ of IRI-TEC are generally smaller than those of the GIM-TEC, especially for $A_4$. The maximum and minimum values of GIM-TEC $A_4$ in Figure 8(g) are 302.27 and −431.47, respectively. The variation range of the IRI-TEC $A_4$ in Figure 8(g) is −138.99 to 165.13. Results in Table 3 indicate that $A_4$ exhibits the largest mean relative bias.

Figure 8 shows that $A_1 - A_6$ reflect the time-varying characteristics of different scales. We conducted EOF decomposition on

$A_1 - A_6$ according to the following equation to divide their diurnal and seasonal variation characteristics:

$$A_i(UT, Doy) = \sum_{k=1}^{N} E_{ik}(UT) \times A_{ik}(Doy) , \tag{13}$$

where $A_i$ represents the coefficient of the $i$ th order the EOF base function. This part is the second-layer EOF decomposition in this study.

Eq. (13) shows that the time-varying feature $E_{ik}$ depending on UT and seasonal variation $A_{ik}$ can be obtained. Given that the

first base function and corresponding coefficient usually demonstrates the main variation, the decomposed first base function $E_{i1}$ and associated coefficient $A_{i1}$ are shown in Figure 9.

The left column of Figure 9 manifests base function $E_{ik}$, which represents the diurnal variation characteristic of the base function $E_i$. The coefficients of the second-layer EOF decomposition $A_{i1}$ represent the variations in long time scales. $A_{i1}$ is shown in the right column of Figure 9. Previous studies have shown that the long time-scale variations of TEC are mainly

influenced by solar and geomagnetic activities and periodical variation. The solar F10.7 index is also shown in the right column of Figure 9 together with $A_{i1}$.

The first base function $E_1$ in Figure 7(a) describes the overall average global TEC, and Figure 9(a) shows $E_{11}$, the diurnal variation characteristic of $E_1$. GIM-TEC and IRI-TEC have similar magnitudes, whereas the diurnal variation of IRI-TEC is insignificant. $A_{11}$ of GIM-TEC and IRI-TEC in Figure 9(b) shows a pronounced semiannual period. However, $A_{11}$ of GIM-TEC

in most days are larger than those of IRI-TEC, and the correlation between F10.7 index and $A_{11}$ of GIM-TEC is evidently observed.

As shown in Figs. 9(c), (e), and (g), the diurnal variations of the second, third, and fourth base functions $E_2 - E_4$ of GIM-TEC and IRI-TEC show minimal discrepancy. Hence, the IRI-2016 model accurately captures the diurnal variations of the solar radiation according to LT and interhemispheric asymmetry.

$A_{21}$ and $A_{31}$ of GIM-TEC and IRI-TEC are shown in Figs. 9(d) and (f). These functions evidently demonstrate a semidiurnal variation period. $A_{21}$ and $A_{31}$ of IRI-TEC during the equinox season are lower than those of GIM-TEC. The correlation between F10.7 index and $A_{21}$ and $A_{31}$ of GIM-TEC is also observed. $A_{41}$ of GIM-TEC and $A_{41}$ of IRI-TEC in Figure 9(h) exhibit an evident annual period variation of interhemispheric asymmetry. However, the summer-to-winter annual variation of GIM-TEC is much larger than that of IRI-TEC.

The fifth and sixth base functions $E_5$ and $E_6$ in Figs. 7(e) and (f) reflect the spatial distribution characteristics along the longitude due to LT. $E_{51}$ and $E_{61}$ in Figs. 9(i) and (k) represent a semidiurnal variation. However, shifts in the peak value time between GIM-TEC and IRI-TEC are detected in $E_{51}$ and $E_{61}$. $A_{51}$ and $A_{61}$ in Figs. 9(j) and (l) exhibit a semiannual variation, and $A_{51}$ and $A_{61}$ of GIM-TEC are relatively consistent with those of IRI-TEC.





We calculated the correlation coefficients between $A_{i1}$ of GIM-TEC and solar F10.7 index. Results are shown in Table 4. Coefficients $A_{11}$, $A_{21}$, and $A_{31}$ are highly related to solar activity.

Table 4. Correlation coefficients between $A_{i1}$ of GIM-TEC and solar F10.7 index

| Coefficient | $A_{11}$ | $A_{21}$ | $A_{31}$ | $A_{41}$ | $A_{51}$ | $A_{61}$ |
|---|---|---|---|---|---|---|
| Correlation coefficient | 0.715 | 0.559 | 0.563 | −0.301 | 0.423 | 0.438 |

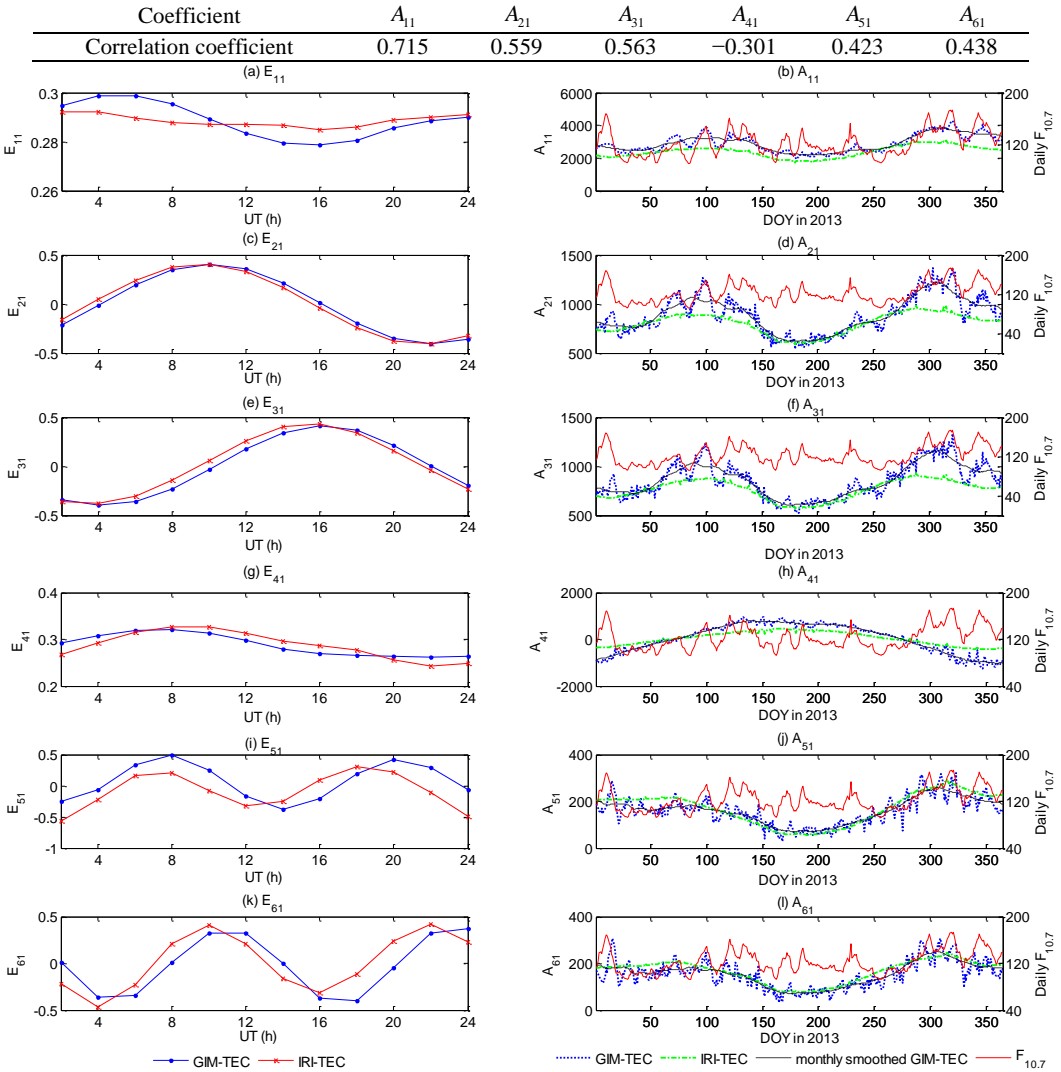

Figure 9. First base function $E_{i1}$ and associated coefficient $A_{i1}$ of the six coefficients $A_1 - A_6$ according to Eq. (12). The monthly smoothed $A_{i1}$ of GIM-TEC and daily solar F10.7 index are shown together with $A_{i1}$.

$A_{11} - A_{61}$ in Figure 9 show that IRI-TEC mainly reflects the annual and semiannual variations of the ionospheric TEC. The monthly and short period variations with solar activity are unrepresented by IRI-TEC.

The IRI-2016 model provides ionospheric parameters of up to 2000 km and will inaccurately predict the TEC up to GNSS satellites located at an altitude of approximately 20,000 km. The IRI-TEC may be smaller than GIM-TEC because of the missing plasmaspheric content.





Despite this situation, $A_{11}$ of IRI-TEC in Figure 9(b) shows a larger underestimation compared with GIM-TEC. The strong correlation between $A_{11}$ of GIM-TEC and solar activity is unrepresented by $A_{11}$ of IRI-TEC. The diurnal variation of the first base function of GIM-TEC represented by $E_{11}$ is partially represented by $E_{11}$ of IRI-TEC. The variance contribution rate of the first EOF component reaches 79.03%; thus, the influence of its coefficient is large for the deviation of IRI-TEC and GIM-TEC.

## 4. Conclusion

In this study, the global TEC prediction performance of the IRI-2016 model was evaluated. The EOF decomposition method was introduced to compare the global TEC data from the IRI-2016 model and GIMs in 2013. The prediction performance of the IRI-2016 model could be evaluated from two perspectives, namely, spatial pattern and temporal variation. The main conclusions are as follows:

1. A general underestimation of the IRI-2016 model can be observed compared with the season averages of global GIM-TEC in 2013, and the RMS of the global TEC deviation is strongly correlated with the solar activity F10.7 index.

2. The six base functions extracted by performing EOF decomposition on the global TEC data from IRI-2016 and GIMs include the following: the variation with the geomagnetic latitude reflecting the daily averaged solar forcing, the diurnal and semidiurnal periodic changes with longitude due to local time, and the interhemispheric asymmetry caused by the annual variation of the inclination angle of the Earth's orbit. The spatiotemporal features extracted from IRI-TEC and GIM-TEC data have good consistency. The IRI-2016 model follows the variation patterns of the observed GIM-TEC.

3. The spatial variation characteristics of IRI-TEC and GIM-TEC can be extracted for comparison on the basis of the same EOF coefficients. Results show that the spatial distribution fluctuation of the IRI-TEC is smaller than that of GIM-TEC. The average relative deviation of the base function representing the interhemispheric asymmetry reaches −56.7%. The interhemispheric asymmetry presents a relatively stable deviation between IRI-TEC and GIM-TEC. The other spatial distribution variations have large deviations in the equator and low latitudes.

4. The temporal variation characteristics of IRI-TEC and GIM-TEC are extracted and compared on the basis of the same EOF base functions. The degree of IRI-TEC changes with time is weaker than that of GIM-TEC. The average relative deviation of the fourth base function coefficient reaches −52.83%. Most diurnal, annual, and semiannual variations of the six base functions of IRI-TEC are consistent with those of GIM-TEC. However, the change with solar activity is unrepresented by IRI-TEC. The diurnal variation of the first base function and the annual variation of the fourth base function have a relatively large deviation between IRI-TEC and GIM-TEC.

5. Results of the spatial and temporal variation characteristic analyses show that the deviation of the first and fourth EOF components between IRI-TEC and GIM-TEC are the two main influencing factors.

*Data availability.* The data used in this study were downloaded from ftp://cddis.gsfc.nasa.gov/ (last accessed: 10 April 2019), http://irimodel.org/ (last accessed: 12 March 2019), and https://omniweb.gsfc.nasa.gov/ (last accessed: 21 April 2019).

*Author contribution.* SL contributed to the conception of the study. SL and JX contributed significantly to the data analysis and manuscript preparation. HZ and JZ performed the model validation and wrote part of the manuscript. ZX and MX contributed to some data analysis work.

*Competing interests.* The authors declare that they have no conflict of interest.

*Acknowledgments.* This work is supported by the Fundamental Research Funds for the Central Universities (Grant Numbers: 2652017105), and the National Natural Science Foundation of China (Grant Number 41574011).



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
