# Peer review of "Global TEC prediction performance assessment of IRI-2016 model based on EOF decomposition"

_Annales Geophysicae, 2019_

## Referee Comment (RC1) · Anonymous Referee #1 · 8 Aug 2019

Review comments on manuscript"**Global TEC prediction performance assessment of IRI-2016 model based on EOF decomposition**" by Li et al., 2019; submitted to Annales Geophysicae

The manuscript compares total electron content from Global Ionospheric Maps products and International Reference Ionosphere during 2013. Empirical Orthogonal Functions are employed to detail the differences between the two datasets. Seasonal average analysis was performed which showed that the IRI model reproduces the equatorial ionisation anomaly distinctively while GIM TEC does show enhancement of TEC over equatorial/low latitude regions, but does not necessarily show the different bands of enhancement at the EIA crests. While related studies exists, I think this work is relevant especially if it clearly shows by how much the IRI under-predicts GIM TEC (in terms of TECu) in different latitude regions. However this is not clearly shown in the current paper.

Additionally, as the authors know, the IRI model provides TEC up to an altitude of 2000 km while GIM TEC products are based on GNSS observations (at about 20000 km). Assuming that the IRI model was 'accurate' at its specified height, it would be missing some plasmaspheric contribution. The authors have missed to point out this important aspect early in the paper. I believe it is related to line 10, page 6, and Figure 1. Information about this is later presented on page 15, line 10.

Below are comments which may assist in improving the paper.

Page 3, line 35: I thought that the GIM TEC products are provided at time resolution of 2 hours. Please cross-check that they are also available for 15 minutes.

Page 4, line 5, please include original references for the hmF2 model options included within the IRI 2016 model. One is based on COSMIC observations (Shubin) and the other one on ionosonde measurements and spheric harmonic method (Altadil).

Page 4, line 10: In the statement "The global TEC date calculated ...". The word 'date' should be data.

Page 5, line 5 is not clear. In the text "If the IRI TEC and GIM TEC are decomposed, then their EOF base functions and coefficients will exhibit poor comparability". Why would this be the case? And do you mean that this would be so, if they were decomposed separately? Assuming that they exhibit some similarities/differences, wouldn't such decomposition bring them out? May be not in magnitude of coefficients or base functions; but perhaps in the trend and identification of physical features?

Following on the previous comment, do you mean that IRI TEC and GIM TEC are combined to form one data file which is later used for decomposition?

Page 7, Table 1, indicate the units of some parameters; maximum, minimum and mean bias; e.g mean bias (TECU).

Page 6, just after line 15: Bias values are computed using IRI TEC and GIM TEC? It is not clear how daily RMS values in 2013 displayed in Figure 2 are computed. Are they just average of the bias values calculated using IRI TEC and GIM TEC?

Page 5, equation 10: Shouldn't RMS be RMSE? This seems to be what is plotted in Figure 2(a). RMSE values of IRI 2016, how are they computed?

Under subsection 3.2: the authors state "We combined the IRI TEC and GIM TEC data ...". If these datasets are combined, how do you obtain Figure 4?

In Figure 3, is global data for 2013 used? How do you account for latitudinal differences? Does this figure reflect only seasonal changes as indicated in the last statement on page 7?

Equation 7 and Figure 7: I am not sure of the physical significance and justification of combining IRI TEC and GIM TEC. Afterall, they have different inherent errors. What can be derived from this combination taken at same grid points can as well be determined from one dataset either GIM TEC or IRI TEC. Otherwise combining these datasets removes the differences/similarities that the authors would want to study? Provide a scientific justification for combining both datasets and what additional features or interpretations are obtained. I don't think that the text in line 15, page 15 is sufficient to justify this inclusion. This has already been discussed.

  Unless I am not understanding equation 7, how do you separately derive A1-A6 for GIM TEC and IRI TEC that you have plotted in Figure 8? Once again, is this necessary? What additional information do we get in Figure 8?

---

## Referee Comment (RC2) · Anonymous Referee #2 · 5 Nov 2019

The paper is very interesting and it is a contribution to IRI-2016 performance, which is always welcome. It uses a statistical technique (EOF) which sometimes is a bit confusing to understand. At least it is my opinion. But overall the paper presents the main differences which are well explained.

I have only some additional comments to those made by Reviewer 1.

Main comments:

In page 5 you mention "Figure 2 demonstrates that the daily predicted RMS of IRI-2016 is in good agreement with the daily solar F10.7 index." If the bias is the deviation from GIM, it is not trivial that it should depend on solar activity level. Why is this?

Which is the data used for Figure 3 ? IRI or GIMS ? I do not understand what this

[Figure]

Figure shows.

Minor correction: At the end of page 3: "University Time (UT)" should be "Universal Time (UT)"

---

## Author Response (AR1)

Manuscript Number: **angeo-2019-79**

Article Title: **Global TEC prediction performance assessment of IRI-2016 model based on EOF decomposition**

5 Dear Editor,

We would like to thank Annales Geophysicae for giving us the opportunity to revise our manuscript. We thank the reviewers for their careful read and thoughtful comments on previous draft. We have carefully taken their comments into consideration in preparing our revision, and hope that the corrections will meet with approval. Revised portion are marked in blue in the marked-up manuscript. The following 10 summarizes how we responded to reviewer comments.

Below is our response to their comments.

Thanks for all the help.

Best wishes.

Yours sincerely,

15 Dr. Shuhui LI

Corresponding Author

**Revision — authors' response**

Reviewer #1:
20 Review comments on manuscript "Global TEC prediction performance assessment of IRI-2016 model based on EOF decomposition" by Li et al., 2019; submitted to Annales Geophysicae
The manuscript compares total electron content from Global Ionospheric Maps products and International Reference Ionosphere during 2013. Empirical Orthogonal Functions are employed to detail the differences between the two datasets.
25 Seasonal average analysis was performed which showed that the IRI model reproduces the equatorial ionisation anomaly distinctively while GIM TEC does show enhancement of TEC over equatorial/low latitude regions, but does not necessarily show the different bands of enhancement at the EIA crests. While related studies exist, I think this work is relevant especially if it clearly shows by how much the IRI under-predicts GIM TEC (in terms of TECu) in different latitude regions. However this is not clearly shown in the
30 current paper.
Answer: According to the reviewer's suggestion, we have added some analysis and discussion about the discrepancies between GIM-TEC and IRI-TEC at different latitudes as follows in the revised manuscript: Considering the different levels of ionospheric activities at different latitudes, mean and RMS values of the discrepancies between seasonal averages of GIM-TEC and IRI-TEC over different latitudinal regions in
35 2013 were calculated. Results are shown in Figure 2. From Figure 2, the mean and RMS values over the area near the equator generally exhibit peak values. GIM-TEC values over the equator and low latitudes are much larger than IRI-TEC values, especially over the ionospheric trough near the magnetic equator shown in Figure 1. The mean and RMS values over Southern Hemisphere during the December solstice are significantly large and also very large over Northern Hemisphere during the June solstice. Therefore, there
40 are large discrepancies between GIM-TEC and IRI-TEC over the summer Hemisphere.

[Figure]

Figure 2. Mean and RMS values of the discrepancies between GIM-TEC and IRI-TEC at different latitudes during four seasons.

Additionally, as the authors know, the IRI model provides TEC up to an altitude of 2000 km while GIM TEC products are based on GNSS observations (at about 20000 km). Assuming that the IRI model was 'accurate' at its specified height, it would be missing some plasmaspheric contribution. The authors have missed to point out this important aspect early in the paper. I believe it is related to line 10, page 6, and Figure 1. Information about this is later presented on page 15, line 10.

Answer: According to the reviewer's instruction, we advanced the relevant paragraph as follows on page 15 to page 6 in the revised manuscript:

"The IRI-2016 model provides ionospheric parameters of up to 2000 km and will inaccurately predict the TEC up to GNSS satellites located at an altitude of approximately 20,000 km. The IRI-TEC may be smaller than GIM-TEC because of the missing plasmaspheric content."

On page 15, we changed the statement as follows:

"Although the IRI-TEC will be smaller than the GIM-TEC because of the missing plasmaspheric content, $A_{11}$ of IRI-TEC in Figure 10(b) shows a quite large underestimation compared with that of GIM-TEC."

Below are comments which may assist in improving the paper.

Page 3, line 35: I thought that the GIM TEC products are provided at time resolution of 2 hours. Please cross-check that they are also available for 15 minutes.

Answer: According to the reviewer's suggestion, we have checked the temporal resolution of IGS GIMs. In terms of temporal resolution, the GIM generated by each IAAC and IGS is different. Final GIMs produced by CODE, ESA, JPL and UPC are provided with a 2h temporal resolution, whereas the CODE-produced IONEX maps are in 1-hour temporal resolution. The temporal resolution of CASG GIMs is 0.5-hour.

In order to describe it more accurately, we changed the expression as follows in the revised manuscript:

"The GIM TEC used in this study is the official IGS combined final product provided by the Crustal Dynamic Data Information System (ftp://cddis.gsfc.nasa.gov). Final GIMs are regular products of the International GNSS Service (IGS) since 1998. These GIMs are provided in the ionosphere exchange format with a spatial resolution of 2.5°×5° in geographic latitude and longitude and a temporal resolution of 2 h."

Page 4, line 5, please include original references for the hmF2 model options included within the IRI 2016 model. One is based on COSMIC observations (Shubin) and the other one on ionosonde measurements and spheric harmonic method (Altadil).

Answer: According to the reviewer's instruction, we have added original references for the hmF2 model options included within the IRI 2016 model in the revised manuscript as follows:

"The recent version of this model is IRI-2016 (Bilitza et al., 2016; Bilitza et al., 2017). After IRI-2012, IRI-2016 exhibits the latest improvement in the model by introducing two new F2 peak height hmF2 modeling

options with their data sources from ionosonde measurements (Altadill et al., 2013) and COSMIC radio occultations (Shubin, 2015)."

Altadill, D., Magdaleno, S., Torta, J. M., Blanch, E.: Global empirical models of the density peak height and of the equivalent scale height for quiet conditions, Adv. Space Res., 52, 1756–1769, https://doi.org/10.1016/j.asr.2012.11.018, 2013.

Shubin, V. N.: Global median model of the F2-layer peak height based on ionospheric radio-occultation and ground-based Digisonde observations, Adv. Space Res., 56, 916–928, https://doi.org/10.1016/j.asr.2015.05.029, 2015.

Page 4, line 10: In the statement "The global TEC date calculated ...". The word 'date' should be data.

Answer: Yes, it is a mistake. We changed "date" to "**data**" in revised version. Thank you.

Page 5, line 5 is not clear. In the text "If the IRI TEC and GIM TEC are decomposed, then their EOF base functions and coefficients will exhibit poor comparability". Why would this be the case? And do you mean that this would be so, if they were decomposed separately? Assuming that they exhibit some similarities/differences, wouldn't such decomposition bring them out? May be not in magnitude of coefficients or base functions; but perhaps in the trend and identification of physical features?

Answer: This sentence is indeed unclear. As you understand, what we want to express is that if they are decomposed separately, it will be difficult to compare in magnitude. We changed the sentence to "If the IRI TEC and GIM TEC are decomposed separately, it is difficult to directly compare their EOF base functions and coefficients in magnitude." in revised version.

The spatial patterns and temporal variations of the global TEC data are separated by EOF decomposition and can be properly represented by the base functions and associated coefficients, respectively. For GIM-TEC data $X_{GIM}$, coefficients $A_{k,\,GIM}$ and EOF base functions $E_{k,GIM}$ will be obtained by using EOF decomposition method. For IRI-TEC data $X_{IRI}$, the coefficients $A_{k,IRI}$ and EOF base functions $E_{k,IRI}$ can be obtained:

$$X_{GIM} = \sum_{k=1}^{N} E_{k,GIM} \cdot A_{k,\,GIM}$$

$$X_{IRI} = \sum_{k=1}^{N} E_{k,IRI} \cdot A_{k,IRI}$$

EOF decomposition will extract main spatial patterns. The six main base functions $E_k$ extracted by performing EOF decomposition on the global TEC from related reference (Talaat and Zhu, 2016) and our study both include the following: the variation with the geomagnetic latitude reflecting the daily averaged solar forcing, the diurnal and semidiurnal periodic changes with longitude due to local time, and the interhemispheric asymmetry caused by the annual variation of the inclination angle of the Earth's orbit.

The spatiotemporal features extracted from IRI-TEC and GIM-TEC data have good consistencies, they are shown in Figs (2) and (3) of this document. Therefore, if GIM-TEC and IRI-TEC are decomposed separately, the results will exhibit obvious similarities in trend and identification of physical features. However, it is not possible to make direct comparisons in magnitude, because $A_{k,\,GIM}$ and $A_{k,IRI}$ are different, $E_{k,GIM}$ and $E_{k,IRI}$ are also different.

So we combined the data to form a whole data set for EOF decomposition and compared the two data sets.

$$\begin{bmatrix} X_{GIM} \\ X_{IRI} \end{bmatrix} = \sum_{k=1}^{N} \begin{bmatrix} E_{k,GIM} \\ E_{k,IRI} \end{bmatrix} \cdot A_k$$

Then, the GIM-TEC and IRI-TEC can be written and reconstruct as follows.

$$X_{GIM} = \sum_{k=1}^{N} E_{k,GIM} \cdot A_k$$

$$X_{IRI} = \sum_{k=1}^{N} E_{k,IRI} \cdot A_k$$

The same coefficients of the EOF base function $A_k$ can be obtained, then $E_{k,GIM}$ and $E_{k,IRI}$ were compared to analyze the difference between GIM-TEC and IRI-TEC. We think the conclusions obtained by the method of this paper are clearer.

Following on the previous comment, do you mean that IRI TEC and GIM TEC are combined to form one data file which is later used for decomposition?

Answer: Yes, IRI TEC and GIM TEC are combined to form one data file which is later used for decomposition.

The two sets of data are arranged in rows or columns as needed.

In our study, We analyzed the global TEC over a 1 year time period (2013) with a 2 h temporal resolution and $37 \times 36 = 1332$ spatial grids, the total epoch number is $12 \times 365 = 4380$. Before EOF analysis, GIM TEC data $X_{GIM}$ should be arranged as follows:

$$X_{GIM} \atop 1332 \times 4380 = \begin{bmatrix} TEC_{grid1,epoch1}^{GIM} & TEC_{grid2,epoch2}^{GIM} & \cdots & TEC_{grid1,epoch4380}^{GIM} \\ TEC_{grid2,epoch1}^{GIM} & TEC_{grid2,epoch2}^{GIM} & \cdots & TEC_{grid2,epoch4380}^{GIM} \\ \vdots & \vdots & \ddots & \vdots \\ TEC_{grid1332,epoch1}^{GIM} & TEC_{grid1332,epoch2}^{GIM} & \cdots & TEC_{grid1332,epoch4380}^{GIM} \end{bmatrix}$$

Coefficients $A_{k,\ GIM}$ and EOF base functions $E_{k,GIM}$ will be obtained by using EOF decomposition method:

$$X_{GIM} = \sum_{k=1}^{N} E_{k,GIM} \cdot A_{k,\ GIM}$$

The same coefficients of the EOF base function, that is, the same time-varying features, can be obtained by arranging IRI-TEC and GIM-TEC according to the same number of columns. That is:

$$\begin{bmatrix} X_{GIM} \\ X_{IRI} \end{bmatrix}_{2664 \times 4380} = \begin{bmatrix} TEC_{grid1,epoch1}^{GIM} & TEC_{grid2,epoch2}^{GIM} & \cdots & TEC_{grid1,epoch4380}^{GIM} \\ TEC_{grid2,epoch1}^{GIM} & TEC_{grid2,epoch2}^{GIM} & \cdots & TEC_{grid2,epoch4380}^{GIM} \\ \vdots & \vdots & \ddots & \vdots \\ TEC_{grid1332,epoch1}^{GIM} & TEC_{grid1332,epoch2}^{GIM} & \cdots & TEC_{grid1332,epoch4380}^{GIM} \\ TEC_{grid1,epoch1}^{IRI} & TEC_{grid1,epoch2}^{IRI} & \cdots & TEC_{grid1,epoch4380}^{IRI} \\ TEC_{grid2,epoch1}^{IRI} & TEC_{grid2,epoch2}^{IRI} & \cdots & TEC_{grid2,epoch4380}^{IRI} \\ \vdots & \vdots & \ddots & \vdots \\ TEC_{grid1332,epoch1}^{IRI} & TEC_{grid1332,epoch2}^{IRI} & \cdots & TEC_{grid1332,epoch4380}^{IRI} \end{bmatrix}$$

Then, we will get EOF decomposition result:

$$\begin{bmatrix} X_{GIM} \\ X_{IRI} \end{bmatrix} = \sum_{k=1}^{N} \begin{bmatrix} E_{k,GIM} \\ E_{k,IRI} \end{bmatrix} \cdot A_k$$

Then, the GIM-TEC and IRI-TEC can be written as follows.

$$X_{GIM} = \sum_{k=1}^{N} E_{k,GIM} \cdot A_k$$

$$X_{IRI} = \sum_{k=1}^{N} E_{k,IRI} \cdot A_k$$

It seems like that we extract common temporal variation factors $A_k$, and we can therefore directly compare the spatial characteristics $E_{k,GIM}$ and $E_{k,IRI}$.

Page 7, Table 1, indicate the units of some parameters; maximum, minimum and mean bias; e.g mean bias (TECU).

Answer: According to the reviewer's suggestion, we added the units of Maximum bias, Minimum bias and Mean bias in Table 1 in the revised manuscript.

Page 6, just after line 15: Bias values are computed using IRI TEC and GIM TEC? It is not clear how daily RMS values in 2013 displayed in Figure 2 are computed. Are they just average of the bias values calculated using IRI TEC and GIM TEC?

Answer: The sentence about how to calculate RMS in line 16, Page 6 is not clear. We changed it as follows in the revised manuscript:

"The gridded values of the global IRI-TEC and GIM-TEC at different UTs for each day of the year 2013 were used to calculate the daily RMS."

The expression of equation (10) in our manuscript is also not clear, so we changed " $RMS = \left[ \sum_{i}^{n} (Y_i - Y_i')^2 / n \right]^{1/2}$ " to " $RMS = \sqrt{\frac{1}{n} \sum_{i}^{n} (Y_i - Y_i')^2}$ " in the revised manuscript.

During a day, the global TEC data has 1322 grid points and 12 epochs. Therefore, both the GIM-TEC data and IRI-TEC data for one day have 1332*12=15984 values.

Daily RMS value in 2013 displayed in Figure 2 is computed by using equation (10):

$$RMS = \sqrt{\frac{1}{n} \sum_{i}^{n} (Y_i - Y_i')^2}$$

Where, n=15984, $Y_i$ and $Y_i'$ are data for GIM-TEC and IRI-TEC respectively.

Page 5, equation 10: Shouldn't RMS be RMSE? This seems to be what is plotted in Figure 2(a). RMSE values of IRI 2016, how are they computed?

Answer: Yes, here is a mistake. RMS and RMSE should be unified. We examined the entire manuscript and used "RMS" in all equations and figures in the revised manuscript.

Under subsection 3.2: the authors state "We combined the IRI TEC and GIM TEC data ...". If these datasets are combined, how do you obtain Figure 4?

Answer: We combined the data as follows:

$$\begin{bmatrix} X_{GIM} \\ X_{IRI} \end{bmatrix}_{2664 \times 4380} = \begin{bmatrix} TEC^{GIM}_{grid1,epoch1} & TEC^{GIM}_{grid2,epoch2} & \cdots & TEC^{GIM}_{grid1,epoch4380} \\ TEC^{GIM}_{grid2,epoch1} & TEC^{GIM}_{grid2,epoch2} & \cdots & TEC^{GIM}_{grid2,epoch4380} \\ \vdots & \vdots & \ddots & \vdots \\ TEC^{GIM}_{grid1332,epoch1} & TEC^{GIM}_{grid1332,epoch2} & \cdots & TEC^{GIM}_{grid1332,epoch4380} \\ TEC^{IRI}_{grid1,epoch1} & TEC^{IRI}_{grid1,epoch2} & \cdots & TEC^{IRI}_{grid1,epoch4380} \\ TEC^{IRI}_{grid2,epoch1} & TEC^{IRI}_{grid2,epoch2} & \cdots & TEC^{IRI}_{grid2,epoch4380} \\ \vdots & \vdots & \ddots & \vdots \\ TEC^{IRI}_{grid1332,epoch1} & TEC^{IRI}_{grid1332,epoch2} & \cdots & TEC^{IRI}_{grid1332,epoch4380} \end{bmatrix}$$

After performing EOF decomposition, we will get:

$$\begin{bmatrix} X_{GIM} \\ X_{IRI} \end{bmatrix}_{2664 \times 4380} = \sum_{k=1}^{N} \underset{2664 \times 1}{E_k} \cdot \underset{4380 \times 1}{A_k} = \sum_{k=1}^{N} \begin{bmatrix} E_{k,GIM} \\ E_{k,IRI} \end{bmatrix}_{2664 \times 1} \cdot \underset{4380 \times 1}{A_k} = \begin{bmatrix} \sum_{k=1}^{N} \underset{1332 \times 1}{E_{k,GIM}} \cdot \underset{4380 \times 1}{A_k} \\ \sum_{k=1}^{N} \underset{1332 \times 1}{E_{k,IRI}} \cdot \underset{4380 \times 1}{A_k} \end{bmatrix}$$

$$\underset{1332 \times 4380}{X_{GIM}} = \sum_{k=1}^{N} \underset{1332 \times 1}{E_{k,GIM}} \cdot \underset{4380 \times 1}{A_k}$$

$$\underset{1332 \times 4380}{X_{IRI}} = \sum_{k=1}^{N} \underset{1332 \times 1}{E_{k,IRI}} \cdot \underset{4380 \times 1}{A_k}$$

That is to say, the two sets of data are arranged together, and after the common coefficients are extracted, the base functions $\underset{2664 \times 1}{E_k}$ are separated to $\begin{bmatrix} E_{k,GIM} \\ E_{k,IRI} \end{bmatrix}$. So we can get two sets of base functions: $\underset{1332 \times 1}{E_{k,GIM}}$ and $\underset{1332 \times 1}{E_{k,IRI}}$, which are shown in Figure 4.

In order to make the expression clearer, we revised equations (6) and (7) in page 5:

$$\text{``}\begin{bmatrix} X_{GIM} \\ X_{IRI} \end{bmatrix} = \sum_{k=1}^{N} \begin{bmatrix} E_{k,GIM} \\ E_{k,IRI} \end{bmatrix} \cdot A_k = \begin{bmatrix} \sum_{k=1}^{N} E_{k,GIM} \cdot A_k \\ \sum_{k=1}^{N} E_{k,IRI} \cdot A_k \end{bmatrix} \tag{6}$$

$$[X_{GIM} \quad X_{IRI}] = \sum_{k=1}^{N} E_k \cdot [A_{k,GIM} \quad A_{k,IRI}] = \left[ \sum_{k=1}^{N} E_k \cdot A_{k,GIM} \quad \sum_{k=1}^{N} E_k \cdot A_{k,IRI} \right] \tag{7''}$$

In Figure 3, is global data for 2013 used? How do you account for latitudinal differences? Does this figure reflect only seasonal changes as indicated in the last statement on page 7?

Answer: Yes, global data for 2013 is used in Figure 3.

The EOF decomposition was conducted on GIM-TEC and IRI-TEC as follow,

$$\begin{bmatrix} X_{GIM} \\ X_{IRI} \end{bmatrix}_{2664\times4380} = \sum_{k=1}^{N} \begin{bmatrix} E_{k,GIM} \\ E_{k,IRI} \end{bmatrix}_{2664\times1} \cdot A_k{}_{4380\times1} = \begin{bmatrix} \sum_{k=1}^{N} E_{k,GIM}{}_{1332\times1} \cdot A_k{}_{4380\times1} \\ \sum_{k=1}^{N} E_{k,IRI}{}_{1332\times1} \cdot A_k{}_{4380\times1} \end{bmatrix}$$

The spatial patterns and temporal variations of the TEC are separated by EOF decomposition and can be properly represented by the base functions and associated coefficients, respectively. $\begin{bmatrix} E_{k,GIM} \\ E_{k,IRI} \end{bmatrix}_{2664\times1}$ represent the TEC's spatial distribution modes, and they are base functions. Six main base functions are shown in Figure 4. And $A_k{}_{4380\times1}$ represents the magnitude of the influence of the $k$th base function component at different epoch. Six coefficients of main base function $A_k{}_{4380\times1}$ are shown in Figure 3.

Therefore, Figure 3 reflects only seasonal changes, while Figure 4 represents the spatial distribution characteristics.

Equation 7 and Figure 7: I am not sure of the physical significance and justification of combining IRI TEC and GIM TEC. Afterall, they have different inherent errors. What can be derived from this combination taken at same grid points can as well be determined from one dataset either GIM TEC or IRI TEC. Otherwise combining these datasets removes the differences/similarities that the authors would want to study? Provide a scientific justification for combining both datasets and what additional features or interpretations are obtained. I don't think that the text in line 15, page 15 is sufficient to justify this inclusion. This has already been discussed.

Answer: After performing EOF decomposition on GIM-TEC and IRI-TEC by using equation (7), we will get base functions $E_k$ and coefficients $A_k$.

$$[X_{GIM} \quad X_{IRI}] = \sum_{k=1}^{N} E_k \cdot [A_{k,GIM} \quad A_{k,IRI}] = \left[ \sum_{k=1}^{N} A_{k,GIM} \cdot E_k \quad \sum_{k=1}^{N} A_{k,IRI} \cdot E_k \right] \tag{7}$$

However, the original TEC data can be reconstructed by using $E_k$ and $A_k$ as follow:

$$X_{GIM} = \sum_{k=1}^{N} A_{k,GIM} \cdot E_k$$

$$X_{IRI} = \sum_{k=1}^{N} A_{k,IRI} \cdot E_k$$

In other words, the EOF decomposition process of equation (7) is reversible. Therefore, decomposition after combining the two sets of data does not lead to errors.

We showed the six main base functions $E_k$ extracted from combined data of IRI TEC and GIM TEC by using Equation (7) in Figure (1). And we also performed EOF decomposition on GIM-TEC and IRI-TEC separately by using Eqs (a) and (b), the base function $E_{k,GIM}$ and $E_{k,IRI}$ are shown in Figs (2) and (3).

$$X_{GIM} = \sum_{k=1}^{N} E_{k,GIM} \cdot A_{k, GIM} \tag{a}$$

$$X_{IRI} = \sum_{k=1}^{N} E_{k,IRI} \cdot A_{k,IRI} \qquad\qquad (b)$$

Although $E_k$ in Figure (1) extracted from combined data is not as same as $E_{k,GIM}$ or $E_{k,IRI}$ in Figs (2) and (3), they do reflect consistent spatial distribution characteristics of global TEC.

Only if common base functions $E_k$ of Equation (7) are used, we can compare $A_{k,GIM}$ and $A_{k,IRI}$ directly. The results will show the difference of the intensity of each base function between GIM-TEC and IRI-TEC.

[Figure]

Figure (1). Base function $E_k$ extracted from combined data of GIM-TEC and IRI-TEC

[Figure]

Figure (2). Base function $E_{k,GIM}$ extracted from GIM-TEC

[Figure]

Figure (3). Base function $E_{k,IRI}$ extracted from IRI-TEC

Unless I am not understanding equation 7, how do you separately derive A1-A6 for GIM TEC and IRI TEC that you have plotted in Figure 8? Once again, is this necessary? What additional information do we get in Figure 8?

Answer: Maybe the equation (7) is not so clear, we have changed it as follows:

$$``[X_{GIM} \quad X_{IRI}] = \sum_{k=1}^{N} E_k \cdot [A_{k,GIM} \quad A_{k,IRI}] = \left[\sum_{k=1}^{N} E_k \cdot A_{k,GIM} \quad \sum_{k=1}^{N} E_k \cdot A_{k,IRI}\right] \qquad (7)"$$

Then, GIM-TEC and IRI-TEC can be written:

$$X_{GIM} = \sum_{k=1}^{N} E_k \cdot A_{k,GIM}$$

$$X_{IRI} = \sum_{k=1}^{N} E_k \cdot A_{k,IRI}$$

Therefore, we can get $A_{k,GIM}$ for GIM-TEC and $A_{k,IRI}$ for IRI-TEC, which are shown in Figure 8. From Figure 8, we can see that the variation of $A_1$ is strongly correlated with solar activity. $A_2$ and $A_3$ have a diurnal variation with UT, and also have a semiannual cycle. $A_4$ has a distinct annual cycle, and $A_5$ and $A_6$ exhibit a semidiurnal cycle and a semiannual cycle. GIM-TEC and IRI-TEC have good consistencies in the above period terms, but the comparison of specific difference in each periodic variation is difficult.

So, we conducted EOF decomposition on $A_1 - A_6$ according to the equation (13) to divide diurnal variation with UT and seasonal variation characteristics of $A_{k,GIM}$ and $A_{k,IRI}$. The results are shown in Figure 9. Therefore, the article continued to discuss the differences between t $A_{k,GIM}$ and $A_{k,IRI}$ based on Figure 9.

We have added some discussion about Figure 8 in the revised manuscript as follows:

"The time-varying characteristics of the coefficients in Figure 8 are very consistent with the results shown in Figure 3. From Figs. 8(a) and (b), the variations of $A_1$ are mainly related to solar activity, and solar activity is the primary determinant of the first base function $E_1$ in Figure 7(a), which describes the overall average of global TEC. From Figs. 8(c)–(f), the EOF coefficients $A_2$ and $A_3$ of GIM-TEC and IRI-TEC all obviously exhibit a diurnal period and a semiannual period. They reflect the diurnal variation of solar radiation change with longitude due to the LT. $A_4$ in Figs. 8(g) and (h) indicate a strong annual cycle variation of the interhemispheric asymmetry of the TEC. $A_5$ and $A_6$ show a semiannual period of the base functions $E_5$ and $E_6$, which represent a longitudinal variation that changes with LT. The EOF coefficients of GIM-TEC and IRI-TEC have consistent annual, semiannual, diurnal, and semidiurnal variations. Therefore, Figure 8 manifests that GIM-TEC and IRI-TEC have highly consistent temporal variation characteristics based on the same spatial distribution modes $E_k$ according to equation (7)."

**Special thanks to you for your good comments and suggestions.**

Reviewer #2:

The paper is very interesting and it is a contribution to IRI-2016 performance, which is always welcome. It uses a statistical technique (EOF) which sometimes is a bit confusing to understand. At least it is my opinion. But overall the paper presents the main differences which are well explained. I have only some additional comments to those made by Reviewer 1.

Main comments:

In page 5 you mention "Figure 2 demonstrates that the daily predicted RMS of IRI-2016 is in good agreement with the daily solar F10.7 index." If the bias is the deviation from GIM, it is not trivial that it should depend on solar activity level. Why is this?

Answer: Fig.2 demonstrates the RMS of bias value of the IRI-TEC and GIM-TEC. Relate research showed that the accuracies of GIM are about 4.0-4.5TECu. Therefore, GIM-TEC data were used as reference values in our study. The ionosphere is ionized by solar radiation, and the correlation coefficient between the global average TEC parameter calculated by GIM and the F10.7 index can reach approximately 0.9. From Fig.2, the ionospheric TEC prediction error of the IRI-2016 model presents a strong correlation with solar activity. We think there are two reasons. On the one hand, when the solar activity is strong, the TEC changes will be more intense. On the other hand, the IRI model does not fully describe the changing characteristics of the ionosphere with solar activity, and this can be verified in the comparison of the later part of the article. In Fig.9, we compared the time variation of IRI-TEC and GIM-TEC based on the same spatial variation component. The solar activity F10.7 index is also given on the figure. The diurnal and semi-diurnal changes of GIM-TEC vary with the F10.7 index, but IRI-TEC values do not reflect this variation characteristics (Figs 9(d), (f), (j), and (l)). The variation of the IRI-TEC is closer to the smoothing effect of the GIM-TEC time variation.

Which is the data used for Figure 3 ? IRI or GIMS ? I do not understand what this Figure shows.

Answer:

The spatial patterns and temporal variations of the global TEC data are separated by EOF decomposition and can be properly represented by the base functions $E_k$ and associated coefficients $A_k$, respectively. We combined the data to form a whole data set for EOF decomposition and compared the two data sets according eq.(6).

$$\begin{bmatrix} X_{GIM} \\ X_{IRI} \end{bmatrix} = \sum_{k=1}^{N} \begin{bmatrix} E_{k,GIM} \\ E_{k,IRI} \end{bmatrix} \cdot A_k \qquad (6)$$

Then, the GIM-TEC and IRI-TEC can be written and reconstruct as follows.

$$X_{GIM} = \sum_{k=1}^{N} E_{k,GIM} \cdot A_k$$

$$X_{IRI} = \sum_{k=1}^{N} E_{k,IRI} \cdot A_k$$

Therefore, the same coefficients of the EOF base function $A_k$ can be obtained, and were shown in Fig. 3. The spatial patterns of GIM-TEC and IRI-TEC $E_{k,GIM}$ and $E_{k,IRI}$ are shown in Fig.4.

This analysis method allows us to clearly see the difference in the spatial variation patterns of the two sets of data.

Minor correction: At the end of page 3: "University Time (UT)" should be "Universal Time (UT)"

Answer: Yes, it is a mistake. We changed " University Time (UT)" to " Universal Time (UT)" in revised version. Thank you.

**Special thanks to you for your good comments and suggestions.**

**List of changes**
**Revised portion are marked in blue in the marked-up manuscript.**

**1. Page 3, Section 2.1**
We changed the expression about GIM TEC as follows:
"The GIM TEC used in this study is the official IGS combined final product provided by the Crustal Dynamic Data Information System (ftp://cddis.gsfc.nasa.gov). Final GIMs are regular products of the International GNSS Service (IGS) since 1998. These GIMs are provided in the ionosphere exchange format with a spatial resolution of 2.5°×5° in geographic latitude and longitude and a temporal resolution of 2 h."

**2. Page 4, Section 2.2**
We have added original references for the hmF2 model options included within the IRI 2016 model in the revised manuscript as follows:
"The recent version of this model is IRI-2016 (Bilitza et al., 2016; Bilitza et al., 2017). After IRI-2012, IRI-2016 exhibits the latest improvement in the model by introducing two new F2 peak height hmF2 modeling options with their data sources from ionosonde measurements (Altadill et al., 2013) and COSMIC radio occultations (Shubin, 2015)."

**3. Page 4, Section 2.2**
We changed "date" to "**data**" in revised version.

**4. Page 4, Section 2.3**
We changed " University Time (UT)" to " **Universal Time (UT)**" in revised version.

**5. Page 5, Section 2.3**
We changed the sentence "If IRI-TEC and GIM-TEC data are decomposed, then their EOF base functions and coefficients will exhibit poor comparability." to:
"If the IRI TEC and GIM TEC are decomposed separately, it is difficult to directly compare their EOF base functions and coefficients in magnitude."

**6. Page 5, Section 2.3**
We have changed Equation (6) from $\begin{bmatrix} X_{GIM} \\ X_{IRI} \end{bmatrix} = \sum_{k=1}^{N} A_k \cdot \begin{bmatrix} E_{k,GIM} \\ E_{k,IRI} \end{bmatrix}$ to:

$$\begin{bmatrix} X_{GIM} \\ X_{IRI} \end{bmatrix} = \sum_{k=1}^{N} \begin{bmatrix} E_{k,GIM} \\ E_{k,IRI} \end{bmatrix} \cdot A_k = \begin{bmatrix} \sum_{k=1}^{N} E_{k,GIM} \cdot A_k \\ \sum_{k=1}^{N} E_{k,IRI} \cdot A_k \end{bmatrix}.$$

**7. Page 5, Section 2.3**
We have changed Equation (7) from $[X_{GIM} \quad X_{IRI}] = \sum_{k=1}^{N} \begin{bmatrix} A_{k,GIM} & A_{k,IRI} \end{bmatrix} \cdot E_k$ to:

$$X_{GIM} \quad X_{IRI}] = \sum_{k=1}^{N} E_k \cdot \begin{bmatrix} A_{k,GIM} & A_{k,IRI} \end{bmatrix} = \begin{bmatrix} \sum_{k=1}^{N} E_k \cdot A_{k,GIM} & \sum_{k=1}^{N} E_k \cdot A_{k,IRI} \end{bmatrix}.$$

**8. Page 5, Section 2.4**
We have changed Equation (10) from $RMS = \left[ \sum_{i}^{n} (Y_i - Y_i')^2 / n \right]^{1/2}$ to:

$$RMS = \sqrt{\frac{1}{n} \sum_{i}^{n} (Y_i - Y_i')^2} \, .$$

**9. Page 6, Section 3.1**
We advanced the relevant paragraph as follows on page 15 to page 6 in the revised manuscript:
"The IRI-2016 model provides ionospheric parameters of up to 2000 km and will inaccurately predict the TEC up to GNSS satellites located at an altitude of approximately 20,000 km. The IRI-TEC may be smaller than GIM-TEC because of the missing plasmaspheric content."

**10. Page 6. Section 3.1**

We have added some analysis and discussion about the discrepancies between GIM-TEC and IRI-TEC at different latitudes as follows in the revised manuscript:

Considering the different levels of ionospheric activities at different latitudes, mean and RMS values of the discrepancies between seasonal averages of GIM-TEC and IRI-TEC over different latitudinal regions in 2013 were calculated. Results are shown in Figure 2. From Figure 2, the mean and RMS values over the area near the equator generally exhibit peak values. GIM-TEC values over the equator and low latitudes are much larger than IRI-TEC values, especially over the ionospheric trough near the magnetic equator shown in Figure 1. The mean and RMS values over Southern Hemisphere during the December solstice are significantly large, and they are also very large over Northern Hemisphere during the June solstice. Therefore, there are large discrepancies between GIM-TEC and IRI-TEC over the summer Hemisphere.

[Figure]

Figure 2. Mean and RMS values of the discrepancies between GIM-TEC and IRI-TEC at different latitudes during four seasons.

Due to the addition of a figure, the numbering of all subsequent pictures has changed.

**11. Page 7, Section 3.1**

We changed the sentence "The bias values between the IRI-TEC and GIM-TEC of all global grid points at different UTs were used to calculate the daily RMS in 2013." to:

"The gridded values of the global IRI-TEC and GIM-TEC at different UTs for each day of the year 2013 were used to calculate the daily RMS."

**12. Page 7, Section 3.1**

We changed "RMSE" to "RMS".

**13. Page 8, Section 3.1**

We added the unit "(TECU)" in Table 1.

**14. Page 9, Section 3.2**

We have modified the labels for the Y axe in Figure 5 (original Figure 4).

**15. Page 11, Section 3.2**

We added a "%" in Table 2.

**16. Page 11, Section 3.2**

We have modified the labels for the Y axe in Figure 6 (original Figure 5).

**17. Page 12, Section 3.3**

We have modified the labels for the Y axe in Figure 7 (original Figure 6).

**18. Page 12, Section 3.3**

[revised manuscript text omitted]

---

## Referee Report (RR1)

Review comments on revised manuscript"**Global TEC prediction performance assessment of IRI-2016 model based on EOF decomposition** " by Li et al., 2019; submitted to Annales Geophysicae

The paper has been revised and the current version is significantly improved. However it still contains some aspects which are not clear and I suggest that the authors look at their suitability as detailed in the following comments:

1. Page 5, line 5; I still think that combining IRI TEC and GIM TEC thereafter decomposing a single data file should be re-looked at. Aren't the authors concerned that by doing this, they are removing the differences/similarities which they intend to study? Magnitude comparison is not a strong justification for combining these datasets. If they exhibit similarities/differences, they will manifest or not show in trends and identified physical features. Therefore this reviewer thinks that IRI TEC and GIM TEC should be decomposed separately.

2. Page 8, section 3.2: This is related to the previous comment. It would have been more straight forward to decompose IRI TEC and GIM TEC separately. There will then be two different figures of Figure 4. This is when the features in Figure 5 can be independently compared. Otherwise, it appears that Figure 5 is generated using values plotted in Figure 4 which were a result of IRI TEC and GIM TEC combination. Please consider re-looking at this. Once this is done, the rest of the figures may slightly change, and perhaps the physical features may remain.
Otherwise, provide a strong justification for combining these datasets not in terms of magnitude. If one was complimenting the other in a different problem, then combining them would perhaps work; but you would need to state the errors associated with these datasets. In the current problem, you are comparing the two datasets and combining them appears to be defeating the intention of the problem.

3. I find the newly added Figure 2 useful. However the physical features are not interpreted. Are these seasonal differences at different latitudes expected? Are there no references in literature to support your observations?

On page 6, in the statement "The IRI-2016 model provides ionospheric parameters of up to 2000 km and will inaccurately predict the TEC up to GNSS satellites located at an altitude of approximately 20,000 km. The IRI-TEC may be smaller than GIM-TEC because of the missing plasmaspheric content"
I suggest changing the words " … and will inaccurately predict ..." to "... is expected to be lower than …" This is because there have been cases where IRI TEC is greater than GIM TEC; and there is sufficient literature showing this.

On page 15 in the statement relating IRI TEC and GIM TEC in terms of A_11; am not sure that associated coefficients, in this case A_11 can be sufficient to explain the magnitude differences between IRI and GIM TEC. Please cross-check and correct if necessary.

---

## Author Response (AR2)

Manuscript Number: **angeo-2019-79**

Article Title: **Global TEC prediction performance assessment of IRI-2016 model based on EOF decomposition**

5  Dear Editor,

We would like to thank Annales Geophysicae for giving us the opportunity to revise our manuscript. We thank the reviewers for their careful read and thoughtful comments on previous draft. We have carefully taken their comments into consideration in preparing our revision, and hope that the corrections will meet with approval. Revised portion are marked in blue in the marked-up manuscript. The following
10  summarizes how we responded to reviewer comments.

Thanks for all the help.

Best wishes.

Yours sincerely,

Dr. Shuhui LI
15  Corresponding Author

**Revision — authors' response**

Reviewer #1:

1 Page 5, line 5; I still think that combining IRI TEC and GIM TEC thereafter decomposing a single data file should be re-looked at. Aren't the authors concerned that by doing this, they are removing the
20  differences/similarities which they intend to study? magnitude comparison is not a strong justification for combining these datasets. If they exhibit similarities/differences, they will manifest or not show in trends and identified physical features. Therefore this reviewer thinks that IRI TEC and GIM TEC should be decomposed separately.

Answer: The reviewer believes that IRI TEC and GIM TEC data should be decomposed by EOF method
25  separately. However, if IRI TEC is decomposed into A * B and GIM TEC is decomposed into C * D, we can analyze similarities and differences between A and C; and we can also discuss differences between the trends of B and C, but we cannot continue more detailed discussion.

That is, after we decomposed IRI TEC and GIM TEC separately, we can get the following pictures:

[Figure]

30  Figure (1). Base function $E_{k,GIM}$ extracted from GIM-TEC

[Figure]

Figure (2). Associated coefficients $A_1 - A_6$ of the first six orders of EOF base functions based $E_{k,GIM}$ extracted from GIM-TEC.

[Figure]

Figure (3). Base function $E_{k,IRI}$ extracted from IRI-TEC

Figure (4). Associated coefficients $A_1 - A_6$ of the first six orders of EOF base functions based $E_{k,GIM}$ extracted from IRI-TEC.

We can only analyze the similarities and differences in spatial patterns by comparing Fig (1) and Fig (3), and discuss the similarities and differences in temporal change characteristics by comparing Fig (2) and Fig (4). The analysis in sections 3.2 and 3.3 later in this article cannot be performed.

In addition, there have been many articles researching EOF decomposition and physical interpretation of GIM TEC (Zhao et al., 2005; Mao et al., 2008; Zhang et al., 2011; Bouya et al., 2012; Zhang et al. , 2013; Uwamahoro and Habarulema, 2015; Talaat and Zhu, 2016; Dabbakuti and Ratnam, 2016, 2017; Chang et al., 2017; Andima et al., 2019; Li et al., 2019). The method of this paper is an effective way to change the perspective to study the problem.We think many of the conclusions we have reached are of certain reference value for understanding the difference between IRI TEC and GIM TEC.

Page 8, section 3.2: This is related to the previous comment. It would have been more straight forward to decompose IRI TEC and GIM TEC separately. There will then be two different figures of Figure 4. This is when the features in Figure 5 can be independently compared. Otherwise, it appears that Figure 5 is generated using values plotted in Figure 4 which were a result of IRI TEC and GIM TEC combination. Please consider re-looking at this. Once this is done, the rest of the figures may slightly change, and perhaps the physical features may remain. Otherwise, provide a strong justification for combining these datasets not in terms of magnitude. If one was complimenting the other in a different problem, then combining them would perhaps work; but you would need to state the errors associated with these datasets. In the current problem, you are comparing the two datasets and combining them appears to be defeating the intention of the problem.

Answer: Yes, after EOF decomposition separately, we will indeed get two different Figures 4, which are Fig. (2) and Fig. (4) in the previous question answer. If IRI TEC is decomposed into A * B and GIM TEC is decomposed into C * D, B and D can be only discussed some differences between their trends.

The basis of our study is that we want to decompose IRI TEC into A * B and GIM TEC into A * E, then we can compare B and E. Figure 4 is the common A, and Figure 5 shows both B and E. Based on Figure 5, you can do a comparative analysis.

In our paper, IRI TEC and GIM TEC are only arranged together. Although the extracted spatial and temporal variation features are not completely consistent with the separate decomposition, the extracted spatial patterns and temporal change features can completely restore the original IRI TEC and GIM TEC data.

In particular, a very similar research method has been widely used in TEC's EOF decomposition. According to previous literature, EOF method can be applied to TEC time series analysis of a single station(Dabbakuti and Ratnam,2017). However, when the EOF method is used for spatio-temporal data, the data at different spatial positions are arranged together and decomposed (Talaat and Zhu, 2016). This does not mean that the data becomes relevant, but the common time-varying characteristics can be extracted from TEC data at different locations.

I find the newly added Figure 2 useful. However the physical features are not interpreted. Are these seasonal differences at different latitudes expected? Are there no references in literature to support your observations?

Answer: According to the reviewer's suggestion, we have added some discussion about the Figure 2 as follows in the revised manuscript:

"From Figure 2, the mean and RMS value over the area near the equator generally exhibit peak values. GIM-TEC values over the equator and low latitudes are much larger than IRI-TEC values, especially over the ionospheric trough near the magnetic equator shown in Figure 1. Due to high solar radiation in the equatorial region and Earth electric and magnetic field, the ionosphere over the equatorial region is at a high ionization level and its changes are complex. There are also anomalies such as equatorial ionization anomaly (EIA) characterized by two low latitude ionization crests of global maximum of plasma densities (Abdu 2016). The IRI model has been reported to overly underestimate the ionospheric TEC at the equatorial station by Shreedevi et al. (2018), and a comparison of IRI model derived TEC and GPS TEC showed a wide departure with ~60% deviation in their study.

The mean and RMS values over Southern Hemisphere during the December solstice are significantly large, and they are also very large over Northern Hemisphere during the June solstice. Therefore, there are large discrepancies between GIM-TEC and IRI-TEC over the summer Hemisphere. The large deviation of the ionospheric TEC estimated by the IRI model in the summer hemisphere indicates that the model cannot

fully reflect the periodic seasonal variation in the ionosphere. As discussed by Li et al. (2016), solar activity component and periodic components are supposed to be the main reasons which account for the difference between the GIMs TEC and the TEC from the IRI-2012 model. However, their conclusions are based on single station time series data. In this article, we will further analyze the IRI model for spatiotemporal data."

We have also added related references in the revised manuscript:

Shreedevi, P. R., Choudhary, R. K., Yadav, S., Thampi, S. and Ajesh, A.:Variation of the TEC at a dip equatorial station, Trivandrum and a mid latitude station, Hanle during the descending phase of the solar cycle 24(2014–2016), J. Atmos. Sol. Terr. Phys., 179, 425-434, 2018.

Abdu, M. A.: Electrodynamics of ionospheric weather over low latitudes, Abdu Geosci. Lett., 3, 11, https://doi.org/10.1186/s40562-016-0043-6, 2016.

On page 6, in the statement "The IRI-2016 model provides ionospheric parameters of up to 2000 km and will inaccurately predict the TEC up to GNSS satellites located at an altitude of approximately 20,000 km. The IRI-TEC may be smaller than GIM-TEC because of the missing plasmaspheric content"

I suggest changing the words " … and will inaccurately predict ..." to "... is expected to be lower than …" This is because there have been cases where IRI TEC is greater than GIM TEC; and there is sufficient literature showing this.

Answer: According to the reviewer's suggestion, we have changed this sentence in the revised manuscirpt as follws:

"The IRI-2016 model provides ionospheric parameters of up to 2000 km and is expected to be lower than the TEC up to GNSS satellites located at an altitude of approximately 20,000 km because of the missing plasmaspheric content."

On page 15 in the statement relating IRI TEC and GIM TEC in terms of A_11; I am not sure that associated coefficients, in this case A_11 can be sufficient to explain the magnitude differences between IRI and GIM TEC. Please cross-check and correct if necessary.

Answer: $A_{11}$ is the first order coefficient of second-layer EOF decomposition by using Eq.(13). From Eq .(5), the effectiveness of the individual EOF components can be quantitatively measured by the ratio of the percentage of the total variance. When we conduct EOF decomposition on data set $A_1$, the variances $r_i$ of different order EOF components are as follows:

| EOF component | $E_{11} \times A_{11}$ | $E_{12} \times A_{12}$ | $E_{13} \times A_{13}$ | $E_{14} \times A_{14}$ | $E_{15} \times A_{15}$ | $E_{16} \times A_{16}$ |
|---|---|---|---|---|---|---|
| Variances $r_i$ | 99.916% | 0.0422% | 0.018% | 0.008% | 0.006% | 0.004% |

From the Table, the first EOF component has already explained 99.916% of the total variance of $A_1$. Furthermore, we showed the original EOF coefficient $A_1$ and its first EOF component $E_{11} \times A_{11}$ in Figure (5). And we can see that they are highly consistent. Therefore, in the second-layer EOF decomposition, the first mode is the most significant, and thus we only present the result for this mode.

[Figure]

Figure (5). The original EOF coefficient $A_i$ and its first EOF components $E_{i1} \times A_{i1}$

Besides, in Chen et al. (2014) and Zhang et al.(2009), researchers implemented a second-layer EOF decomposition and only the first mode was studied to explore the temporal variations explicitly. In order to make the manuscript clearer, we added some explanation in the revised manuscript as follows: "According to the percentage variance of the second-layer EOF decomposition, the first EOF component has already explained more than 99% of the total variance of $A_i$. Therefore, the first EOF component is the most significant, and we will only present the first order result of the second-layer EOF decomposition in this study."

**Special thanks to you for your good comments and suggestions.**

**List of changes**
**Revised portion are marked in blue in the marked-up manuscript.**

1.   Page 6, Section 3.1

5 We changed the expression about IRI-2016 model as follows:

[revised manuscript text omitted]